# Surgery for Liver Metastasis of Non-Colorectal and Non-Neuroendocrine Tumors

**DOI:** 10.3390/jcm11071906

**Published:** 2022-03-29

**Authors:** Shadi Katou, Franziska Schmid, Carolina Silveira, Lina Schäfer, Tizian Naim, Felix Becker, Sonia Radunz, Mazen A. Juratli, Leon Louis Seifert, Hauke Heinzow, Benjamin Struecker, Andreas Pascher, M. Haluk Morgul

**Affiliations:** 1Department for General, Visceral and Transplant Surgery, University Hospital Muenster, 48149 Muenster, Germany; shadi.katou@ukmuenster.de (S.K.); franziska_schmid96@yahoo.de (F.S.); c_silv01@uni-muenster.de (C.S.); linamariaschafer@gmail.com (L.S.); tizianamir.naim@ukmuenster.de (T.N.); felix.becker@ukmuenster.de (F.B.); sonia.raduenz@ukmuenster.de (S.R.); mazen.juratli@ukmuenster.de (M.A.J.); benjamin.struecker@ukmuenster.de (B.S.); andreas.pascher@ukmuenster.de (A.P.); 2Department of Gastroenterology and Hepatology, University Hospital Muenster, 48149 Muenster, Germany; leonlouis.seifert@ukmuenster.de (L.L.S.); h.heinzow@bk-trier.de (H.H.); 3Department of Internal Medicine I, Krankenhaus der Barmherzigen Brüder Trier, 54292 Trier, Germany

**Keywords:** liver metastases, colorectal liver metastases, non-colorectal and non-neuroendocrine liver metastases, liver resection

## Abstract

Surgery has become well established for patients with colorectal and neuroendocrine liver metastases. However, the value of this procedure in non-colorectal and non-neuroendocrine metastases (NCRNNELMs) remains unclear. We analyzed the outcomes of patients that underwent liver surgery for NCRNNELMs and for colorectal liver metastases (CRLMs) between 2012 and 2017 at our institution. Prognostic factors of overall and recurrence-free survival were analyzed, and a comparison of survival between two groups was performed. Seventy-three patients (30 NCRNNELM and 43 CRLM) were included in this study. Although the mean age, extrahepatic metastases, and rate of reoperation were significantly different between the groups, recurrence-free survival was comparable. The 5-year overall survival rates were 38% for NCRNNELM and 55% for CRLM. In univariate analysis, a patient age of ≥60 years, endodermal origin of the primary tumor, and major complications were negative prognostic factors. Resection for NCRNNELM showed comparable results to resection for CRLM. Age, the embryological origin of the primary tumor, and the number of metastases might be the criteria for patient selection.

## 1. Introduction

Surgical treatment of colorectal liver metastasis (CRLM) has been well established over the past few decades. Multimodal approaches, innovative surgical techniques, and interdisciplinary therapy concepts have contributed to achieving better long-term survival rates in patients who were previously deemed palliative. In fact, recent analyses suggest 5- and 10-year survival rates of 40–58% and 12–36%, respectively, when radical resection (R0) was achieved [1,2,3,4,5,6,7,8]. In parallel to the implementation of surgical strategies for CRLM, hepatic neuroendocrine metastases have also become a field of interest. While hepatic metastases that originate from neuroendocrine tumors generally indicate a negative prognosis, resection of the metastases has been shown to be beneficial for patients in both palliative and curative settings of the disease with a favorable influence on long-term outcomes [9,10,11].

Since colorectal cancers and neuroendocrine tumors mainly metastasize through the portal vein, the incidence of their metastasis in the liver is high. Nevertheless, the liver is a common metastatic site of various other primary tumors. The role of resection in such settings has been scarcely explored until recently, mainly due to the rarity of these diseases compared to CRLM [12,13,14]. An epidemiologic study from the Netherlands showed that 46% of adenocarcinoma liver metastases were colorectal in origin, whereas gastric, pancreatic, or esophageal metastases represented only 15%. Metastasis of breast cancer is the most common metastatic disease of a non-splanchnic organ, accounting for only 8.2% of all metastases [15].

Recent developments in surgical techniques, resulting in reduced postoperative morbidity following liver surgery, have increased the courage to offer hepatic resection to patients with non-colorectal non-endocrine liver metastases (NCRNNELMs). Thus, the indication for hepatic resection for NCRNNELM has to be redefined in-line with these developments. However, there is a current gap in knowledge regarding surgical outcomes, as most available data on this subject lack follow-up results of hepatic re-section and dismiss the comparison with CRLM.

The aim of this study was to analyze the outcomes of patients treated for liver metastasis of NCRNNE origin compared with patients suffering from CRLM and to identify prognostic factors of overall and recurrence-free survival in this cohort.

## 2. Materials and Methods

This study was carried out as an observational retrospective single-center trial, which analyzed all patients that underwent surgical resection for liver metastases between 2012 and 2017 at our tertiary center. The study was conducted in accordance with the Declaration of Helsinki and approved by our local ethics committee (ID: ID 2019-636-f-S). Due to the retrospective character of the analysis, patient consent was waived. Patient data were collected from hospital archives and electronic patient records.

The inclusion criteria were histologically proven liver metastases and surgical treatment of the metastases with curative intention. The exclusion criteria were age <18 years, palliative resections, and other tumor control therapies for liver metastasis, such as ablation, radiation, or embolization (Figure 1).

All patients were discussed at our local interdisciplinary tumor board prior to oncological procedures. Due to their unique biology, liver metastases of neuroendocrine tissues were excluded from our study. Patients with extrahepatic disease manifestations (beyond the primary lesion) were not excluded. Resection of four or more segments in one session was considered as major hepatectomy. Oligometastatic liver disease was defined as less than five metastases [16]. Synchronous metastases were those that were diagnosed within the first six months of primary diagnosis [17].

Two groups were defined based on the site of the primary tumor: the CRLM and NCRNNELM groups. In addition to the patients’ demographic data, information regarding the primary disease, description of hepatic lesions, and surgical procedure was acquired. The patients’ postoperative course was screened for complications and follow-up. The tumor burden score was calculated according to the formula A2 + B2 = C2 (A: maximum tumor diameter; B: number of tumors; C: Tumor Burden Score) [18]. To overcome the heterogeneity of primary cancers, tumors were categorized according to embryological origin: ectodermal, mesodermal, and endodermal.

### Statistics

Statistical analysis was performed using SPSS (V. XX, IBM, Armonk, New York, NY, USA). Data were described as the mean and standard deviation or median and range. Paired and unpaired Student’s *t*-tests were carried out for comparison of parameters, as appropriate. For multivariate analysis and group comparison, log-rank and Cox regression analyses were performed. Overall survival (OS) was calculated from the day of surgery to death or last follow-up. Recurrence-free survival (RFS) was calculated from the day of surgery to the first diagnosis of recurrence if radical resection of the liver and extrahepatic manifestations were initially achieved. Analyses of OS and RFS were obtained by using the Kaplan–Meier method. A *p*-value < 0.05 was considered statistically significant.

## 3. Results

### 3.1. Characteristic Data

A total of 73 patients were included in this study, of which 43 had CRLM and 30 had NCRNNELM (Figure 1). The median follow-up time of all patients after hepatic resection was 45 months.

Descriptive data of both groups are presented in Table 1. Female patients represented 60.5% and 50% of CRLM and NCRNNELM cases, respectively. The patients’ mean age in the NCRNNELM group was 54.3 years, which was significantly younger than that in the CRLM group (64.6 years, *p* = 0.03). Extrahepatic manifestations and metastases were significantly more frequent in patients with NCRNNELMs than those with CRLM (5% vs. 30%, *p* = 0.003). There was no significant difference regarding sex, American Society of Anasthesiologists (ASA) state, pre-existing liver conditions, or synchronicity of metastases between the two groups. Solitary metastasis in the NCRNNELM group was more common (55% vs. 80%, whereas cases of oligo- and multiple metastases were more frequent in the CRLM group (30% vs. 16.7% and 15% vs. 3.3%, respectively, *p* = 0.02). However, the tumor burden score was similar in both groups (4.5 for CRLM; 3.9 NCRNNELM). The majority of lesions were smaller than 5 cm and located in the right lobe of the liver in both groups. However, bilobar lesions were more frequent in CRLM (25.6% vs. 6.7%, *p* = 0.03). Although a major hepatectomy was more commonly performed for CRLM, the operation time and extent of liver resection did not differ significantly between the groups. Margin-free resection (R0) was achieved in 93% and 90% of the CRLM and NCRNNELM groups, respectively (*p* = 0.67). None of the patients in the NCRNNELM group required reoperation or died in the first 30 days after surgery, whereas seven patients (16.3%) with CRLM underwent reoperation due to complications after liver re-section, of which one patient (2.3%) died on postoperative day 14 after extended right hepatectomy due to portal vein thrombosis and liver failure. Hence, the major complication rate in CRLM was higher than in the NCRNNELM group (Clavien–Dindo ≤ 3a were 37.2% and 46.7%, Clavien–Dindo > 3a were 18.6% and 6.7%, respectively).

Although the overall survival (OS) of patients after resection of CRLM was higher than that of the NCRNNELM group, this difference was not statistically significant (Figure 2). However, the 1- and 3-year survival rates were significantly higher in the CRLM group (93% vs. 60%, *p* = 0.001; 72% vs. 43%, *p* = 0.01). There was no significant difference in 5-year survival between the CRLM and NCRNNELM groups (55% vs. 38%, *p* = 0.26). In the CRLM group there were more patients having more than one liver lesion, but the tumor burden was similar in both groups. On the contrary, extrahapatic disease was more common in the NCRNNELM group. However, none of these parameters were significantly predictive of overall survival in NCRNNELM patients.

To estimate recurrence-free survival, we excluded patients with R1 resection and untreated extrahepatic disease. Of the remaining 22 patients in the NCRNNELM group, 12 patients (54.5%) developed disease recurrence during follow-up; in seven cases (25.9%), hepatic recurrence was reported, and in five cases (18.5%), extrahepatic disease recurrence was reported. The mean recurrence-free survival was 11.7 months. In the CRLM group, three patients were excluded due to R1 resection, and of the remaining 40, disease recurrence was reported in 29 cases (72.5%). There was no significant difference in recurrence-free survival between the two groups; however, after 1 year, patients with CRLM tended to develop recurrence more frequently than those with NCRNNELM (Table 2, Figure 2).

### 3.2. Predictive Factors of Overall Survival and Recurrence-Free Survival in NCRNNELM

Primary tumors for NCRNNCELM (*n* = 30) were pancreatic adenocarcinoma (*n* = 4), renal cell carcinoma (*n* = 3), esophageal cancer (*n* = 3), gastrointestinal stromal tumor (*n* = 3), melanoma (*n* = 3), sarcoma (*n* = 3), testicular cancer (*n* = 3), papillary adenocarcinoma (*n* = 2), ovarian cancer (*n* = 2), thyroid cancer (*n* = 2), breast cancer (*n* = 1), and gallbladder cancer (*n* = 1). Accordingly, metastases were of mesodermal, ectodermal, and endodermal origin in 46.7%, 13.3%, and 40% of cases, respectively. Overall survival (OS) in this group at 1, 3, and 5 years was 60%, 43.3%, and 38%, respectively, although it should be noted that 5-year survival was not applicable in patients treated after 2015. In univariate analysis, a patient age over 60 years of age and a primary tumor of endodermal origin were identified as negative prognostic factors for OS; however, none of these factors proved significant in multivariate analysis as independent factors (Table 3, Figure 3). Recurrence-free survival (RFS) rates at 1, 3, and 5 years were 47.4%, 38.9%, and 23.5%, respectively, with a median RFS of 25.1 ± 28.9 (0–99) months. Embryology of the primary tumor stood out as a significant predictor for RFS in univariate and multivariate analyses, with endodermal origin demonstrating the poorest prognosis. Univariate analysis confirmed the number of liver lesions as a significant factor for RFS; however, the multivariate analyses on ASA, age, extrahepatic manifestation, synchronicity, location of metastases, size of the largest lesion, adjuvant or neoadjuvant chemotherapy, and postoperative complications did not show any significance.

## 4. Discussion

To date, the role of radical surgical treatment of NCRNNELM is a topic of ongoing debate. Currently, many patients with NCRNNELM are treated as palliative despite modern advances in liver surgery and improvement of multimodal therapy concepts. Our study analyzed the 5-year survival of patients after liver resection for NCRNNELM and compared them to a cohort of surgically treated patients with CRLM. In addition, we focused on prognostic predictors in NCRNNE liver metastasis patients.

Adam et al. shed light in 2006 on resection of NCRNNELM in a multicenter study including 1452 patients and developed an algorithm to assist selecting patients and predicting their outcomes [19]. This was a milestone in this field, yet a decade later, some of those factors might be outdated [20]. In their population, they identified an age of over 60 years, extrahepatic metastases, and major hepatectomy as negative prognostic factors. In a more recent study with 100 patients, Holzner et al. found residual disease, female sex, endodermal origin, and onset of metastatic disease within 24 months of primary diagnosis to have a negative prognostic effect on outcome [21]. However, in their data, they excluded patients with extrahepatic or extra-abdominal disease and selected only patients with “curative” intent surgery. We explicitly did not exclude patients with extrahepatic disease in our cohort and found no negative correlation with either OS or RFS. We were only able to reproduce two of the previously suggested negative prognostic factors on OS in our results: patient age over 60 years and endodermal origin of the primary tumor. On the other hand, the number of metastases in addition to the origin of the primary tumor was found to be a further prognostic factor of recurrence-free survival.

There have been several previous publications on the surgical treatment of various metastatic liver diseases, many of which reported breast or genitourinary cancer as the most common primary [22,23,24]. In particular, metastases of genitourinary primaries have shown a more favorable outcome in comparison to those of the gastrointestinal tract, and a median survival time as much as three times longer has been described [25,26]. In our cohort, the most common primary site was the gastrointestinal tract (33.3%). Fewer metastases of genitourinary primaries were observed, and only one case of breast cancer was included. The contrast with some other studies was due to different geography and distribution of primary disease, which might have had an effect on different outcomes. Wakabayashi et al. showed in a recent multicentric analysis a 5-year survival rate of 41% in 205 patients after curative resection of non-colorectal liver metastases of the stomach and pancreas as the most common primary sites, which was similar to our results [27]. 

The establishment of liver resection in CRLM has come a long way, and the initial results 20 years ago on surgical treatment of CRLM showed comparable results to the most recent data on surgical resection of NCRNNELM cases [28]. Therefore, one must assume that there is room for improvement in this field. In particular, the implementation and improvement of minimally invasive liver resection for NCRNNELM might lead to further improvement in this area as it did in surgical treatment of CRLM [29,30]. Furthermore, minimally invasive treatment of the primary tumor would lead to a better postoperative performance score of patients. Thus, several patients would be suitable for additive surgery in terms of liver metastasis [31]. On the other hand, in the last decade, significant progress in the multidisciplinary treatment of oncological diseases was seen. Thus, neoadjuvant concepts have been widely investigated in gastrointestinal malignancies, showing encouraging results for tumor shrinkage and improved survival [32,33]. 

In our cohort, the liver resection of the metastasis of tumors with endodermal origin showed a significantly worse prognosis. The sample size would not allow the comparison of all the entities. However, i.e., the liver metastases of pancreatic adenocarcinoma were only solitary tumors. In one patient, the resection of the metastasis was conducted simultaneously during the primary operation for pancreatic cancer. Three patients underwent hepatic surgery following primary pancreas surgery and additive chemotherapy. Recently, Shao et al. showed the feasibility of simultaneous pancreas and liver resection in the oligometastatic concept due to pancreatic cancer, and it showed a significant benefit for the patients following resection and (neo)adjuvant chemotherapy and surgery in comparison to patients that underwent palliative regimens only [34]. Although they included nine patients without neoadjuvant chemotherapy due to the intraoperative diagnosis of liver lesions, they favored chemotherapy for oligometastatic diseases. We also propose that the NCRNELM should be rapidly evaluated for aggressive systemic therapy, since there are encouraging steps in the oncological treatment of, especially, pancreatic [33], esophageal, gastric [35], and renal cell cancer [36]. Taking all these developments into consideration, it is more likely that more patients with liver metastasis of NCRNNE could be candidates for hepatic resection.

Although our data were not able to detect independent prognostic factors for outcomes after liver resection of NCRNNELM, as demonstrated in previous studies, we found that age and embryological origin of primary tumors had an effect in univariate analysis. This was probably due to the relatively small sample of patients in our study. Moreover, although the short-term survival of patients after liver resection for CRLM was better, the overall and 5-year survival results showed no significant difference between the CRLM and NCRNNELM groups. Thus, we were not able to identify any factors that significantly impaired overall survival following surgery on NCRNNELM. This finding was in line with the recent publications of Patkar et al. and Lok et al. with similar sample sizes [37,38]. This suggested that patients with NCRNNELM could benefit from radical treatment in the long run. Therefore, patients with NCRNNELM should be evaluated for surgical treatment in terms of the concept of oligometastases, before palliative regimens are introduced. 

As a retrospective study with a small number of patients, due to the rarity of such surgically treated cases, even in a high-volume single center, our data had certain limitations. Primary diseases were heterogeneous in our cohort, and comparisons of those diseases were mostly carried out in other studies according to histology. We deliberately did not categorize subgroups depending on histology or site of primary tumor because the number of patients in each subgroup would have been too small for statistical comparison. Instead, the embryonic origin of the primary tumor was considered. Furthermore, outcome data beyond 5 years were missing in this study, since we only analyzed patients treated between 2012 and 2017.

Since the occurrence of liver surgery for NCRNNELM is rare, attempts have to be made to build collaborations to achieve bigger cohorts. However, the indication, surgical strategy, and the treatments in terms of adjuvant and neoadjuvant chemotherapy for NCRNNELM may differ between centers, thus it still would be challenging to define the objective criteria for NCRNNELM. On the other hand, as stated by a Dutch group recently, even in a nationwide data analysis, variation on outcomes following liver surgery can occur [39]. 

## 5. Conclusions

Despite the heterogeneous distribution of the primary disease, our results concluded that hepatic resection of NCRNNELM might be feasible for patients under 60 years of age and with metastasis of non-endodermal primaries, and showing satisfying 5-year survival results. For recurrence-free survival, multiple metastases and endodermal origin of the primary tumor appeared to have an unfavorable influence. However, in multivariate analyses, our data did not identify any significant factor that affected overall survival. Hence, cases of NCRNNELM should be individually discussed by multidisciplinary boards with an experienced liver surgeon, and surgical treatment should be considered. To establish a treatment algorithm for these patients, further prospective and multicentric studies are needed.

## Figures and Tables

**Figure 1 jcm-11-01906-f001:**
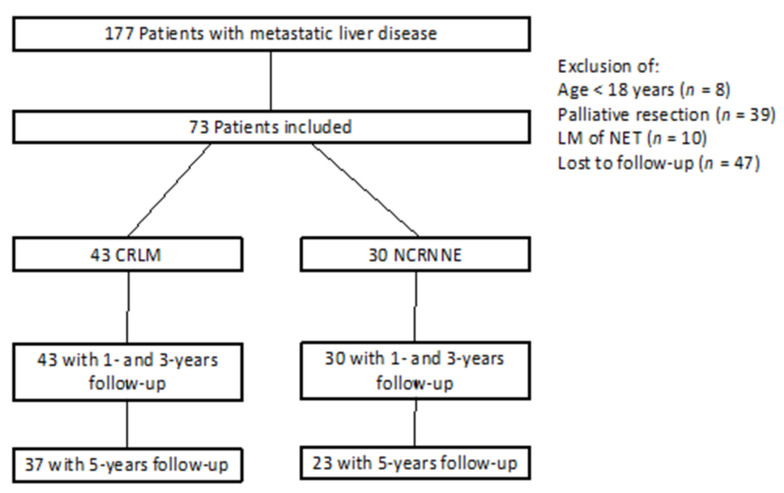
Flow chart of patient inclusion.

**Figure 2 jcm-11-01906-f002:**
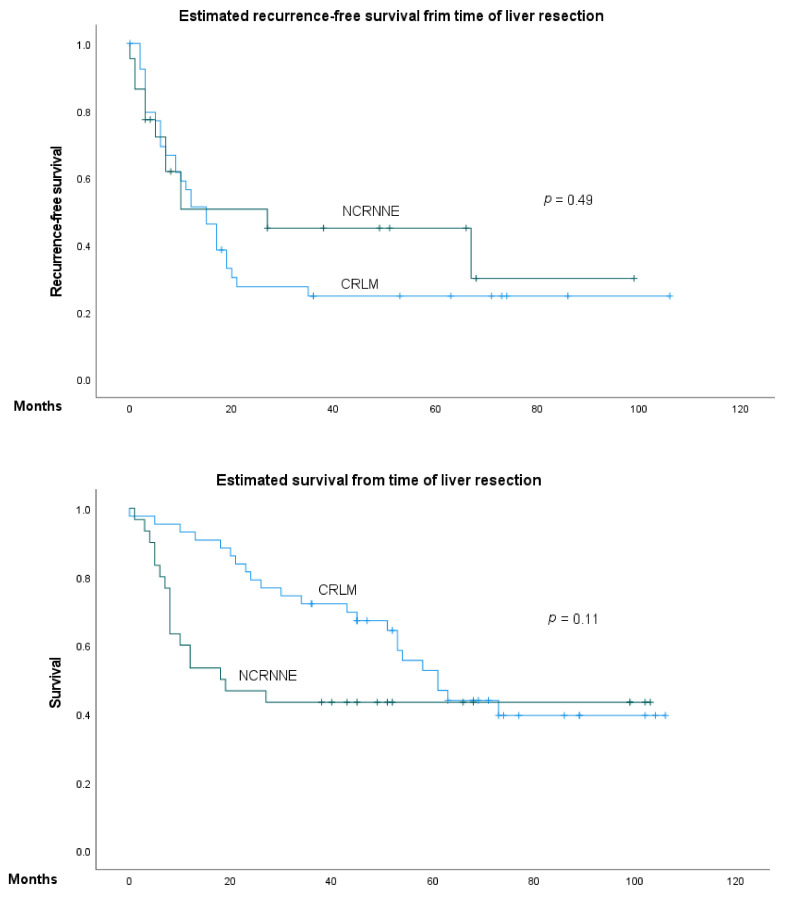
Kaplan–Meier analysis of the overall and recurrence-free survival of patients with NCRNNE and CRLM.

**Figure 3 jcm-11-01906-f003:**
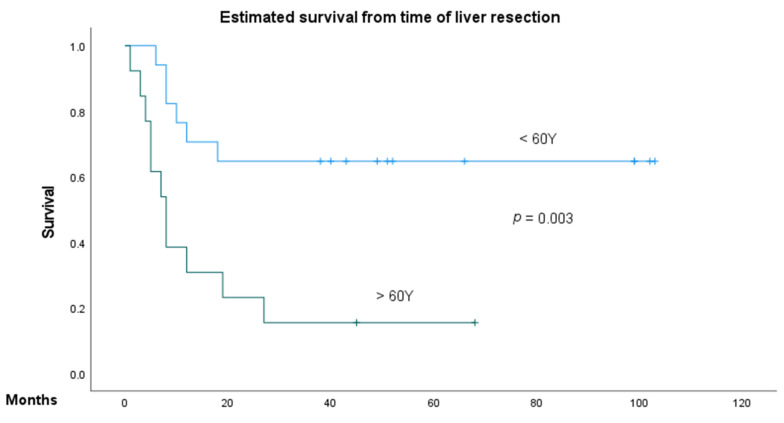
Kaplan–Meier analysis of the age (above and below 60 years) and embryological origin of the primary tumor on overall survival of the patient with NCRNNE.

**Table 1 jcm-11-01906-t001:** Patient characteristics.

Parameter	CRLM *n* = 43	NCRNNE *n* = 30	*p* (<0.05)
All patients	43 (100.0)	30 (100.0)	
Age (median and range)	64.5 (35–90)	54.3 (20–80)	0.03 ^a^
Sex (*n*, %)			
Male	26 (60.5)	15 (50.0)	n.s.
Female	17 (39.5)	15 (50.0)	n.s.
BMI (kg/m^2^, mean ± SD)	27.2	26.0	n.s.
ASA score (*n*, %)			
≤2	28 (65.1) *	22 (73.3)	n.s.
>2	12 (27.9)	8 (26.7)	
Synchronicity (*n*, %)			
synchronous	19 (44.2)	12 (40.0)	n.s.
metachronous	24 (55.8)	18 (60.0)	n.s.
Extrahepatic metastasis (*n*, %)			
Yes	2 (4.7)	9 (30.0)	0.003 ^b^
No	41 (95.3)	21 (70.0)	
Number of metastases (*n*, %)			
Solitary	22 (51.1) *	24 (80.0)	
Oligo	12 (27.9)	5 (16.7)	0.02 ^b^
Multiple	6 (13.9)	1 (3.3)	
Size of biggest lesion (*n*, %)			
≤5 cm	36 (83.7) *	25 (83.3)	n.s.
>5 cm	4 (9.3)	5 (16.7)	
TBS (mean ± SD)	4.5 ± 4.6	3.9 ± 3.4	n.s.
Location (*n*, %)			
Right lobe	25 (58.1)	17 (56.7)	
Left lobe	7 (16.3)	11 (36.7)	0.03
Bilobar	11 (25.6)	2 (6.7)	
Preoperative chemotherapy (*n*, %)			
Yes	28 (65.1)	17 (56.7)	n.s.
No	15 (34.9)	13 (43.3)	
Postoperative chemotherapy (*n*, %)			
Yes	21 (48.8) *	12 (40.0) *	n.s.
No	20 (46.5)	16 (53.3)	
Liver resection (*n*, %)			
minor	36 (83.7)	29 (96.7)	n.s.
major	7 (16.3)	1 (3.3)	
Surgery time (min, mean ± SD)	214	237.8	n.s.
ICU (day, mean ± SD)	5.6	4.1	n.s.
Blood Transfusion (*n*, %)			
Yes	7 (16.2) *	2 (6.66) *	n.s.
No	34 (79.0)	26 (86.6)	
R-status (*n*, %)			
R0	40 (93.0)	27 (90.0)	n.s.
R1	3 (7.0)	3 (10.0)	
Reoperation (*n*, %)			
Yes	7 (16.3)	0 (0)	0.02 ^b^
No	36 (83.7)	30 (100.0)	
Complications (*n*, %)			
none	18 (41.8) *	14 (46.7)	
CD ≤ 3a	16 (37.2)	14 (46.7)	n.s.
CD > 3a	8 (18.6)	2 (6.7)	
ICU readmission (*n*, %)			
yes	2 (5.0)	1 (3.3)	n.s.
no	41 (95.0)	29 (96.7)	

CRLMs: colorectal liver metastases, NCRNNE: non-colorectal non-neuroendocrine, ASA: American Society of Anesthesiologists, BMI: body mass index, TBS: tumor burden score, ICU: intensive care unit, CD: Clavien–Dindo score, n.s.: not signifcant, ^a^: Student’s *t*-test, ^b^: Fischer’s exact test, * missing patients’ data.

**Table 2 jcm-11-01906-t002:** Parameters on recurrence-free survival.

Variables	Recurrence-Free Survival					
	Univariate			Multivariante		
	HR	95% CI	*p*	HR	95% CI	*p*
ASA, ≤2 vs. >2	1.22	0.32–4.66	0.76			
Sex, male vs. female	0.56	0.16–1.89	0.35			
Age, ≤60 years vs. >60 years	1.36	0.41–4.45	0.60			
Primary embryology, mesoderm vs. ectoderm vs. endoderm	3.41	1.34–8.67	0.01	2.96	1.10–7.94	0.03
Extra-hepatic disease manifestation, yes vs. No	0.76	0.16–3.55	0.73			
Synchronicity, synchronous vs. metachronous	1.44	0.43–4.81	0.55			
Timing of metastases, ≤24 months vs. >24 months	0.99	0.29–3.35	0.99			
Number of metastases, solitary vs. Multiple	12.59	2.09–75.74	0.006	5.71	0.92–35.47	0.06
Location of metastases, right vs. Left vs. Bilobar	0.94	0.31–2.85	0.92			
Size of biggest lesion, ≤5 cm vs. >5 cm	0.26	0.31–2.21	0.21			
Neoadjuvant chemotherapy, yes vs. no	0.56	0.16–1.92	0.36			
Adjuvant chemotherapy, yes vs. no	0.75	0.19–2.83	0.67			
Clavien-Dindo, 0 vs. ≤3a vs. >3a	1.67	0.62–4.44	0.30			

ASA; American Society of Anesthesiologists, HR; hazard ratio, CI; confidence interval.

**Table 3 jcm-11-01906-t003:** Parameters on overall survival.

Variables	Overall Survival					
	Univariate			Multivariate		
	HR	95% CI	*p*	HR	95% CI	*p*
ASA, <2 vs. >2	1.76	0.65–4.77	0.26			
Sex, male vs. female	0.74	0.28–1.92	0.53			
Age, ≤60 years vs. >60 years	4.03	1.48–10.99	0.006	1.90	0.51–7.01	0.33
Primary embryology, mesoderm vs. ectoderm vs. endoderm	2.46	1.37–4.41	0.003	1.93	0.91–4.08	0.08
Extra-hepatic disease manifestation, yes vs. no	0.88	0.30–2.50	0.81			
Synchronicity, synchronous vs. metachronous	0.80	0.30–2.12	0.66			
Timing of metastases, ≤24 months vs. >24 months	1.86	0.65–5.30	0.24			
Number of metastases, solitary vs. multiple	1.42	0.46–4.39	0.53			
Location of metastases, right vs. Left vs. bilobar	0.96	0.43–2.14	0.93			
Size of biggest lesion, <5 cm vs. >5 cm	1.02	0.29–3.56	0.97			
Neoadjuvant chemotherapy, yes vs. No	0.54	0.20–1.40	0.20			
Adjuvant chemotherapy, yes vs. No	0.65	0.23–1.79	0.40			
R status, R0 vs. R1	1.66	0.37–7.28	0.50			
Blood transfusion, yes vs. No	4.63	0.99–21.52	0.05			
Clavien-Dindo, 0 vs. ≤3a vs. >3a	1.83	0.89–3.78	0.09			

ASA: American Society of Anesthesiologists, HR: hazard ratio, CI: confidence interval.

## Data Availability

The data presented in this study are available on request from the corresponding author. The data are not publicly available due to the approval of the ethic committee.

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
