# Peer review of "Surgery for Liver Metastasis of Non-Colorectal and Non-Neuroendocrine Tumors"

_jcm, 2022, doi:10.3390/jcm11071906_

Round 1

Reviewer 1 Report

Line 58-59: The aim of this study was to analyze the outcomes of patients treated for liver metas- 58 tasis of NCRNNE origin compared with patients suffering from CRLM and to identify 59 prognostic factors of overall and recurrence-free survival in this cohort.

Comment: Would it be more meaningful to compare the outcomes of patients who did not get resection of their liver mets, than to compare against those with colorectal cancer? It doesn't make sense to compare outcomes of different types of cancer. 

Lines 75-77: Resection of four or more segments in one session 75 was considered major hepatectomy. Oligometastatic liver disease was defined as less than 76 five metastases. Synchronous metastases were those that were diagnosed within the first 77 six months of primary diagnosis. 

Suggestion: Could you please include references for these definitions? 

Lines 82-84: The patient’s postoperative course was screened for complications and follow-up. 82 The Tumor Burden Score was calculated according to the formula A2+B2=C2 (A: maxi- 83 mum tumor diameter; B: number of tumors; C: Tumor Burden Score).

Suggestion: Reference required. 

Lines 271-273: Conclusions.  

Comments: There is already a body of evidence present for liver mets resection for many of the primary tumours included in this study - including gastric, breast, pancreas, GIST, ovarian cancers. It is difficult to draw any conclusions about liver met resection since the primary tumours are so heterogenous, and this study doesn't add to the current knowledge. 

Author Response

1)         Line 58-59: The aim of this study was to analyze the outcomes of patients treated for liver metastasis of NCRNNE origin compared with patients suffering from CRLM and to identify prognostic factors of overall and recurrence-free survival in this cohort.

Comment: Would it be more meaningful to compare the outcomes of patients who did not get resection of their liver mets, than to compare against those with colorectal cancer? It doesn't make sense to compare outcomes of different types of cancer.

We thank the reviewer for this valuable comment and fully agree that the relatively small sample size to many subgroups are the limitations of our study. We also agree that a comparison of the surgical treatment and oncological treatment of metastatic diseases would be a very valuable tool. However, as the reviewer might also agree, this would have more subgroups (different etiologies, different chemotherapy regimen, likely more extrahepatic disease etc.).  

The establishment of liver resection in CRLM has come a long way, and the initial results 20 years ago on surgical treatment of CRLM showed comparable results to most recent data on surgical resection of NCRNNELM cases. We postulate that to overrule the cancer, the only way is to understand the biology of it and act interdisciplinary, as we do for colorectal cancer.  

2)        Lines 75-77: Resection of four or more segments in one session 75 was considered major hepatectomy. Oligometastatic liver disease was defined as less than 76 five metastases. Synchronous metastases were those that were diagnosed within the first 77 six months of primary diagnosis.

Suggestion: Could you please include references for these definitions?

We thank the reviewer for highlighting the definition of both oligometastatic disease and Synchronous metastasis. We added the references to the manuscript.

3)        Lines 82-84: The patient’s postoperative course was screened for complications and follow-up. 82 The Tumor Burden Score was calculated according to the formula A2+B2=C2 (A: maxi- 83 mum tumor diameter; B: number of tumors; C: Tumor Burden Score).

Suggestion: Reference required.

We thank the reviewer for highlighting the definition of TBS. We added the reference to the manuscript.

4) Lines 271-273: Conclusions. 

Comments: There is already a body of evidence present for liver mets resection for many of the primary tumours included in this study - including gastric, breast, pancreas, GIST, ovarian cancers. It is difficult to draw any conclusions about liver met resection since the primary tumours are so heterogenous, and this study doesn't add to the current knowledge.  

We thank the reviewer for this valuable comment and fully agree that the relatively small sample size (especially in the mentioned subcategories) is a limitation of our study. We also adapted the title of our manuscript according to this issue. However, we think that our study adds valuable data to the literature since the metastatic disease of some carcinomas are widely accepted for palliative chemotherapy. This is almost where we stood 20 years ago with colorectal carcinoma. In our study we could show  that especially patients younger than 60 years old with a carcinoma of ectodermal origin might benefit from the surgery.

Reviewer 2 Report

This is a retrospective study looking at the efficacy of surgery for non-colorectal, non-neuroendocrine liver metastases. Indeed, because of rarity of such situations, I understand that robust results cannot be derived from the present cohort.

The authors clarified that origin of primary lesion seems to be associated with survival outcomes. However, because of the small number of series, it would be difficult to perform reliable analysis including multivariate analysis. Unfortunately, given the potential selection bias and too small number of cohort compared to the reported multicenter studies, it would be difficult to answer to the clinical question presented in the title of the manuscript.

Author Response

1)         This is a retrospective study looking at the efficacy of surgery for non-colorectal, non-neuroendocrine liver metastases. Indeed, because of rarity of such situations, I understand that robust results cannot be derived from the present cohort.

The authors clarified that origin of primary lesion seems to be associated with survival outcomes. However, because of the small number of series, it would be difficult to perform reliable analysis including multivariate analysis. Unfortunately, given the potential selection bias and too small number of cohort compared to the reported multicenter studies, it would be difficult to answer to the clinical question presented in the title of the manuscript.

We thank the reviewer for this valuable comment and fully agree that the relatively small sample size (especially in the mentioned subcategories) is a limitation of our study. We therefore adapted the title of our manuscript.

Round 2

Reviewer 2 Report

Modification of the title seems to be reasonable as the authors revised in the present version.
Although the manuscript is well written including discussion on the limitation points, there seem to be no supportive data for the last sentence in the abstract. I would recommend rewording this part.

Author Response

We thank the reviewer for this important suggest, we now revised the abstract to point out the correct conclusion.